# Localization of False Data Injection Attack in Smart Grids Based on SSA-CNN

**Kelei Shen, Wenxu Yan \*, Hongyu Ni and Jie Chu**

School of Internet of Things, Jiangnan University, Wuxi 214000, China
\* Correspondence: ywx01@jiangnan.edu.cn

**Abstract:** In recent years, smart grids have integrated information and communication technologies into power networks, which brings new network security issues. Among the existing cyberattacks, the false data injection attack (FDIA) compromises state estimation in smart grids by injecting false data into the meter measurements, which adversely affects the smart grids. Current studies on FDIAs mainly focus on the detection of its existence, but there are few studies on its localization. Most attack localization methods have difficulty locating the specific bus or line that is under attack quickly and accurately, have high computational complexity and are difficult to apply to large power networks. Therefore, this paper proposes a localization method for FDIAs that is based on a convolutional neural network and optimized with a sparrow search algorithm (SSA–CNN). Based on the physical meaning of measurement vectors, the proposed method can precisely locate a specific bus or line with relatively low computational complexity. To address the difficulty of selecting hyperparameters in the CNN, which leads to the degradation of localization accuracy, a SSA is used to optimize the hyperparameters of the CNN so that the hyperparameters are optimal when using the model for localization. Finally, simulation experiments are conducted on IEEE14-bus and IEEE118-bus test systems, and the simulation results show that the method proposed in this paper has a high localization accuracy and can largely reduce the false-alarm rate.

**Keywords:** false data injection attack (FDIA); localization; sparrow search algorithm; convolutional neural network

## 1. Introduction

In recent years, the rapid development of computer, communication and automatic control technologies has led to the gradual integration of information technology and physical systems, resulting in the emergence of cyber–physical systems (CPSs). As one of the typical CPSs, smart grids are more intelligent and efficient than the traditional power systems. The distributed interconnection of the power system equipment is realized with the help of an information system, and a new energy architecture is built [1]. However, with the continuous integration of physical systems and information technology, smart grids are subjected to an increasing number of network attacks that not only damage the information system itself but may also affect the operation of the physical systems.

The false data injection attack was first proposed in 2009; it is a data integrity network attack, and its main purpose is to compromise the integrity and consistency of data [2,3]. In power systems, attackers add attack vectors to the measurement data through certain paths, and the carefully designed attack vectors are able to evade the bad data detection (BDD) algorithms in the energy management system (EMS). Since the state estimation module of the smart grid is in the supervisory control and data acquisition (SCADA) system, the attack vectors influence the state estimation by utilizing the tampered measurement data to undermine the security and stability of the power systems for illegal profit.

The impact of FDIAs on smart grids mainly lies in economy and stability. Through a well-designed FDIA, attackers can seriously affect the normal operation of the power

market, especially the marginal price of position (LMP). Attackers manipulate the LMP on a large scale by tampering with real-time measurements [4,5]. Attackers manipulate the data in the power grids or its own electricity meters with the FDIA, thereby reducing the electricity cost, obtaining profits in a disguised way and destroying the normal operation of the power market [6]. Through a FDIA, attackers can disrupt the stable operation of a smart grid, causing great losses to the country and people. In 2015, the SCADA system in Ukraine experienced a cyberattack, and the hackers caused widespread and prolonged power outages throughout Ukraine by malicious attacks, affecting the normal life of millions of people, which was the first case of a large-scale power outage caused by the failure of the physical grid due to a cyberattack [7].

A FDIA has the characteristics of good concealment, high flexibility, wide attack range and strong destructive power, so many researchers have conducted research on FDIAs; however, most of the research has only focused on the detection of FDIAs, ignoring the problem of localization for FDIAs [8–13]. In order to reduce the impact of a FDIA on a smart grid and prevent an invisible FDIA that is lurking in a power system from causing secondary damage to the system or power equipment, it becomes an urgent need to accurately locate the attacked sensor bus or line after detecting a FDIA. In [14], the minimum variance untraced Kalman filter (MVUKF) and the deviation of WLS were used for the detection of a FDIA. The power system was divided into several subsystems, and the hypothesis test was applied to each subsystem for the localization of the FDIA. The subsystems with the FDIA were further divided into smaller subsystems, and the localization was implemented using the same method. This method has two shortcomings: first, the subsystem division and the threshold setting are difficult, and the steps are complicated, which make it difficult to apply to large systems; and second, the localization of a FDIA using this method is only able to locate the subsystem, not the specific buses or lines being attacked. Interval observers were designed in [15,16] for state estimation of the power system to achieve global detection and used a logical localization matrix for the localization and isolation of the FDIA. However, the simulation only compared the proposed interval observer with the ordinary Kalman filter, which cannot illustrate the advantages of the designed observer. The logical localization matrix in the paper was based on the idea of partitioning for localization, and the simulation was based on a five-bus subsystem, which required four steps to locate the specific buses or lines being attacked; moreover, the calculation was complicated and not applicable to large power systems. An adaptive hierarchical network attack detection and localization framework was proposed in [17] for distributed active power distribution systems. The author used an improved grid-partitioning method based on spectral clustering to divide a large power network into several sub-regions and then used a method based on signal anomaly strength to detect the localization of network attacks. A specific score was evaluated for all possible attacked localizations within the sub-regions, and the region with the highest score was identified as being attacked; however, sometimes this method only determined the range rather than the specific buss or lines.

In practice, complex power systems are the majority, so the observer-based localization methods for FDIAs have the problems of difficult threshold setting and high computational complexity. The data-driven approaches do not have those problems because they do not require a system model and have very excellent performance for the detection and localization of FDIAs based on data sets [18]. Interaction variables and LSTM were used in [19] for the detection and localization of FDIAs. However, the paper used a five-bus system simulation, which limited the application of this method to large-scale power systems, and the LSTM model was built for each quantity measurement in the paper, which would greatly increase the complexity. A fault and attack localization and classification system was proposed in [20] to classify and locate cyberattacks on devices and communication networks. A multi-label classifier based on DNN was used to identify true data and false data. Multiple types of faults and network attacks were classified, and it was able to identify 10 faults and eight attacks and locate them accurately. However, in the simulation, there

were only 209 data points for each attack type. Compared to detecting FDIAs, identifying attacked areas has been less studied. Ref. [21] proposed a graph neural network (GNN) model based on an autoregressive moving average type graph filter. The authors treated the localization of the FDIA as a multi-label classification task and realized the classification using the proposed model. In [22], autoencoders were used to detect large-scale FDIAs. The authors generated normal volume measurement vectors based on generative adversarial networks and then compared the difference between the attacked vectors and normal vectors with an algorithm to achieve localization. The method proposed in the article had a high detection rate, but the localization accuracy was low and prone to false localization. Hoffding adaptive trees were used in [23] to classify contingencies, such as equipment failures and network attacks, in power systems with high classification accuracy.

Most of the above methods have high complexity and have performance that is limited by the size of the test systems; in addition, the selection of hyperparameters in the deep learning-based localization methods is difficult, and improper selection would lead to the degradation of localization accuracy. In this paper, the localization problem of FDIAs in smart grids is considered as a multi-label classification problem, and we use a convolutional neural network (CNN) as a multi-label classifier to distinguish the attacked measurements from normal measurements. At the same time, considering the problem of poor results obtained by manually selecting neural network hyperparameters, this paper proposes a method that uses a convolutional neural network optimized with a sparrow search algorithm (SSA–CNN) for localization of FDIAs in smart grids. The structure of this paper is organized as follows. First, the models of power systems and FDIAs are analyzed. Second, the sparrow search algorithm and convolutional neural network are introduced, and the research of the localization of FDIAs in smart grids is conducted. Finally, simulation experiments are conducted in IEEE14-bus and IEEE118-bus test systems to demonstrate the effectiveness and accuracy of the proposed method.

The contributions of this paper are outlined as follows:

1.  A data-driven localization method for FDIAs in smart grids is proposed, which uses a CNN classifier to locate FDIAs and the attacked buses and lines.
2.  A novel CNN structure with measurement vectors as input and classification vectors as output is designed. A SSA is used to optimize multiple parameters to obtain the CNN model with optimal localization effect. At the same time, the CNN model will change according to the different optimization results and thus improve the localization accuracy.
3.  To the best of our knowledge, this article is the first to develop the SSA–CNN to locate FDIAs in smart grids. The proposed method is verified on the measurement data set containing invisible FDIAs and compared with other advanced localization methods for FDIAs. The results show that the proposed method has the highest localization accuracy.

## 2. Models

### 2.1. Power System Model

When the network topology and various parameters of the power system are certain, the measurement equation based on AC nonlinearity can be written as

$$z = h(x) + v \tag{1}$$

where $z$ and $x$ denote the measurement vector and state vector of the power system, respectively, $v$ denotes the measurement noise, and $h(x)$ denotes the measurement function.

The measurement vector $z$ is obtained from the sensor as shown in the equation

$$z = (P_i, Q_i, P_{ij}, Q_{ij})^T \tag{2}$$

where $P_i$ and $Q_i$ denote the active power and reactive power at bus $i$, respectively, and $P_{ij}$ and $Q_{ij}$ denote the active current and reactive current of branch $ij$, respectively.

The state vector $x$ represents the real-time state of the power system and consists of the voltage magnitude of each bus and voltage phase angle of each bus excluding the balance bus as shown in the equation

$$x = (v_i, \theta_i)^T \tag{3}$$

where $v_i$ denotes the voltage magnitude at bus $i$, and $\theta$ denotes the voltage phase angle at the bus $i$ (the balanced bus voltage phase angle is 0).

The measurement function $h(x)$ represents the nonlinear relationship between $x$ and $z$, whose exact form is determined with the grid structure and linear parameters and is calculated from the active/reactive power, active/reactive currents and voltage in the electrical line topology [24], which is as follows

$$h(x) = \begin{bmatrix} P_i(v_i, v_j, \theta_i, \theta_j) \\ Q_i(v_i, v_j, \theta_i, \theta_j) \\ P_{ij}(v_i, v_j, \theta_i, \theta_j) \\ Q_{ij}(v_i, v_j, \theta_i, \theta_j) \\ v_i \end{bmatrix} \tag{4}$$

The specific expressions of the items in $h(x)$ are shown as

$$P_i(v_i, v_j, \theta_i, \theta_j) = \sum v_i v_j (G_{ij} \cos \theta_{ij} + B_{ij} \sin \theta_{ij}) \tag{5}$$

$$Q_i(v_i, v_j, \theta_i, \theta_j) = \sum v_i v_j (G_{ij} \sin \theta_{ij} - B_{ij} \cos \theta_{ij}) \tag{6}$$

$$P_{ij}(v_i, v_j, \theta_i, \theta_j) = v_i^2 g_{ij} - v_i v_j (g_{ij} \cos \theta_{ij} + b_{ij} \sin \theta_{ij}) \tag{7}$$

$$Q_{ij}(v_i, v_j, \theta_i, \theta_j) = -v_i^2 (b_{ij} + y_c) - v_i v_j (g_{ij} \sin \theta_{ij} - b_{ij} \cos \theta_{ij}) \tag{8}$$

where $\theta_{ij} = \theta_i - \theta_j$ denotes the phase angle difference between bus $i$ and bus $j$; $G_{ij}$ denotes the real part of the element in row $i$, column $j$ of the conductance matrix; $B_{ij}$ denotes the imaginary part of the element in row $i$, column $j$ of the conductance matrix; $g_{ij}$ denotes the conductance of the branch $ij$; $b_{ij}$ denotes the conductance of the branch $ij$; and $y_c$ denotes the line to ground conductance.

### 2.2. False Data Injection Attack Model

In real power grids, bad data detection (BDD) algorithms are used to detect sampling errors on the measurement instruments. The traditional BDD algorithms use the largest normalized residual (LNR) test hypothesis to detect bad data by computing the 2-parametric number of measurement error quantities. When the state vectors are independent of each other, the measurement vectors conform to a normal distribution, and the residuals obey the chi-square distribution, then the BDD algorithms can be formulated as

$$\|r\| = \|z - h(x)\| < \varepsilon \tag{9}$$

where $\varepsilon$ is the detection threshold of the BDD algorithm.

Since the premise of a false data injection attack on the power system is that the attacker knows the topology and relevant data of the network, the attacker modifies the measurement data of the important buses to cause the state estimation results to deviate from the true value. Therefore, it is assumed that the attacker's modified measurement vector is $z_a$, which satisfies the equation

$$z_a = z + a \tag{10}$$

where $a$ denotes the attack vector carefully designed by the attacker with the same dimensionality as the measurement vector, which contains non-zero elements, indicating erroneous data that can tamper with the state volume using the state estimation process.

Since the state vector after the state estimation can be tampered with using false data, let the state vector after the false data attack be $x^*$, which satisfies the equation

$$x^* = x + c \tag{11}$$

where $c$ denotes the difference between the original state vector and the tampered state vector.

The conventional bad data detection algorithms can detect the bad data in the volume measurement, and the bad data are the deviations from the normal data that are not generated by human factors but by the noise generated by the sensor acquisition, data transmission and other processes. When the attacked measurement vector is detected with the BDD algorithms, we can obtain

$$\|r_a\| = \|z_a - h(x)\| = \|z + a - h(x^*)\| = \|z + a - h(x + c)\| \tag{12}$$

The common FDIA cannot evade the BDD algorithms of smart grids, and this kind of FDIA is called a "random FDIA". With the development of technology, there is a new kind of FDIA whose attack vector is specially constructed to evade the residual test of the BDD algorithms; this kind of FDIA is called an "invisible FDIA". When the attacker constructs the attack vector as $a = h(x + c) - h(x)$, the residual detection expression can be obtained as

$$\|r_a\| = \|z_a - h(x)\| = \|z + a - h(x + c)\| = \|z - h(x)\| = \|r\| \tag{13}$$

In this case, the residual test results before and after the attack are the same, and the constructed attack vector is able to evade the BDD algorithms [25].

## 3. Localization of a False Data Injection Attack Based on SSA–CNN

A CNN is usually used to classify images, and its ability to better preserve the semantic structure of images renders the classification results more accurate [26]. The volume measurement data of power systems are one-dimensional, time-series data, and using CNN classifiers can identify the differences between the attacked measurements and normal measurements, which allows accurate localization. The accuracy of localization using a CNN is closely related to the hyperparameter values of the CNN, which are usually selected manually, but this cannot guarantee the accuracy of the localization results. A SSA has great advantages over other optimization algorithms in terms of convergence, merit-seeking ability and stability [27]; therefore, in this paper, a SSA is used to optimize the CNN to obtain the optimal hyperparameters and thus improve the accuracy of the localization method.

### 3.1. The CNN Model

A typical CNN model consists of an input layer, several convolutional layers, several pooling layers, a flatten layer, a fully connected layer and an output layer. The filter of the convolutional layer performs feature extraction with sliding steps and outputs high-level features to the fully connected layer after processing by the pooling and flattening layers to achieve regression prediction or classification.

Since the power system volume measurements are time-series data, this paper uses a one-dimensional convolutional layer. Figure 1 shows the CNN structure used in this paper, which is divided into six parts as follows:

(1) Input layer

The input data consists of the power system measurement vector $z$, which is a one-dimensional column vector.

(2) Convolutional layer

The feature extraction of input measurement is performed by the convolutional layer, and the process can be expressed as

$$x_k^j = f\left(\sum_{i=1}^{N}(x_{i,k}^{j-1} * W_{i,k}^j) + b_k^j\right) \tag{14}$$

where $x_k^j$ denotes the output of the convolutional layer $j$, $N$ denotes the number of vectors of the current convolutional layer, $x_{i,k}^{j-1}$ denotes the input of the current convolutional layer, $W_{i,k}^j$ denotes the weight of each input, and $b_k^j$ denotes the bias.

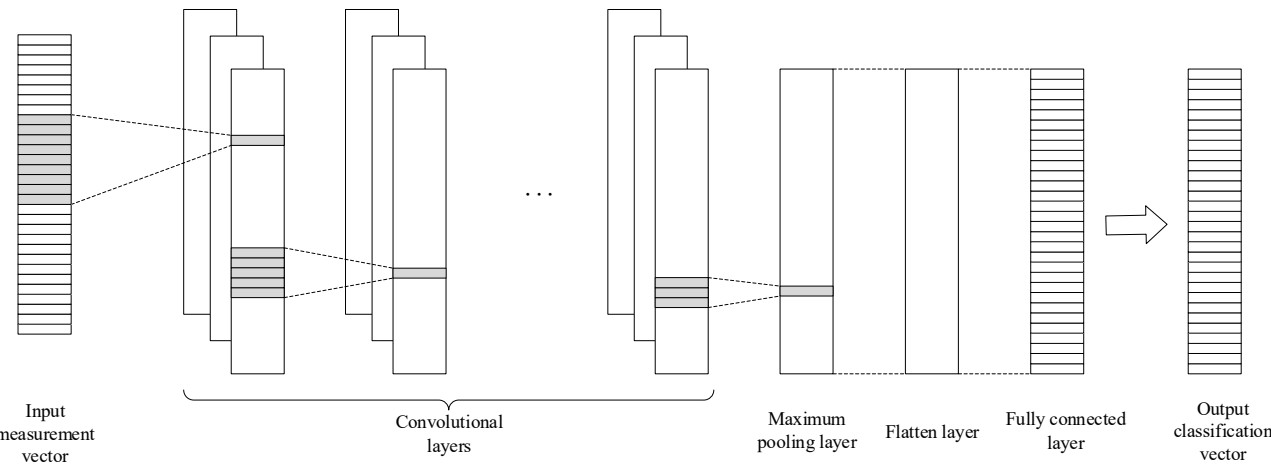

**Figure 1.** Structure of the proposed CNN.

The activation function of the convolutional layer is the rectified linear unit (ReLU). The reasons are as follows:

(1) Using a function, such as sigmoid, is computationally intensive, while using ReLU saves a significant amount of computational effort.
(2) For deep networks, when the sigmoid function is backpropagated, the gradient will easily disappear, resulting in training failure of the deep network.
(3) ReLU causes the output of some neurons to be 0. This reduces the interdependence of the parameters, which alleviates the overfitting problem.

The expression of the ReLU activation function is shown as

$$f(x) = \max(0, x) \tag{15}$$

(3) Pooling layer

In this paper, the maximum pooling layer can be used to extract the high-level features of the sequence and reduce the computational effort; the maximum value used is the output

$$\hat{x}_k^j = \max(x_k^j, \ldots, x_{k+r-1}^j) \tag{16}$$

where $\hat{x}_k^j$ denotes the maximum value in vector $x_k^j$ to $x_{k+r-1}^j$, and $r$ denotes the step size of the pooling layer.

(4) Flatten layer

The flatten layer integrates the feature vectors processed by the convolutional layer and the maximum pooling layer and transports them to the fully connected layer, which acts as a takeover.

(5) Fully connected layer

The fully connected layer uses the sigmoid function with the equation

$$f(x) = \frac{1}{1 + \exp(-x)} \tag{17}$$

The sigmoid function maps the output to the interval between $(0, 1)$, enabling the output to be used for a binary classification task with smooth, easy-to-derive characteristics.

(6) Output layer

The measurement vector $z$ passes through the whole convolutional neural network, turning into the output classification vector $y_k$. Since the sigmoid function is used in the fully connected layer, the elements in the classification vector $y_k$ are mapped to the interval $(0, 1)$. At this point, each element in $y_k$ is classified as 0 or 1 with the logical judgment formula as

$$y_k^i = \begin{cases} 0 & y_k^i \leq \varepsilon \\ 1 & y_k^i > \varepsilon \end{cases} \tag{18}$$

where $y_k^i$ denotes the first element of the classification vector $y_k$, and $\varepsilon$ denotes the predefined threshold value.

### 3.2. Sparrow Search Algorithm

A sparrow search algorithm is proposed based on the foraging behavior of sparrows, which can be divided into two categories of individual sparrows: explorers and followers. According to [28], there are six rules of a SSA.

(1) In the whole population, explorers are energy-rich and are responsible for searching areas with sufficient food and providing foraging areas and directions for followers. The high-energy reserve in the algorithm is related to the fitness value of individual sparrows.

(2) The individual sparrows will send an alert signal when they find natural enemies (predators) and, when the alert value is greater than the safe value, the explorers and the followers will enter the safe area to forage.

(3) The identities of the explorers and followers can be interchanged, but the proportion of individual sparrows in the whole population will not change between the two identities.

(4) Sparrows with higher energy reserves will act as explorers. In order to obtain more energy, the lower energy followers may fly to other places to forage for food.

(5) During foraging, the followers will always be able to follow the explorers with higher energy reserves to forage. In order to improve their predation rate, they will spy on the explorers and thus compete for more resources.

(6) When there is a threat, the sparrows at the edge of the group will move to the safe area, while the sparrows in the middle of the group will move randomly.

In the simulation experiments, virtual sparrows are needed for the food search, and the population of sparrows can be represented as

$$X = \begin{bmatrix} x_1^1 & x_1^2 & \cdots & x_1^d \\ x_2^1 & x_2^2 & \cdots & x_2^d \\ \cdots & \cdots & \cdots & \cdots \\ x_n^1 & x_n^2 & \cdots & x_n^d \end{bmatrix} \tag{19}$$

where $X$ denotes the sparrow population; $x$ denotes the individual sparrow; $d$ denotes the dimension of the population, numerically the same as the CNN parameters to be optimized; and $n$ denotes the number of sparrows.

The fitness value is shown as

$$F_X = \begin{bmatrix} f([\ x_1^1 & x_1^2 & \cdots & x_1^d\ ]) \\ f([\ x_2^1 & x_2^2 & \cdots & x_2^d\ ]) \\ \cdots \\ f([\ x_n^1 & x_n^2 & \cdots & x_n^d\ ]) \end{bmatrix} \tag{20}$$

where $F_X$ denotes the fitness matrix; and $f$ denotes the fitness value, which is obtained by calculating the root mean squared error (RMSE) of the predicted and state values.

In the sparrow search algorithm, individuals with higher fitness values will have priority in obtaining food and, from rule 1, it is known that explorers will obtain a larger

foraging search range than followers. From rule 2, the position update of the explorer during each iteration is described as

$$X_{i,j}^{t+1} = \begin{cases} X_{i,j}^t \cdot \exp(\frac{-i}{\alpha \cdot iter_{\max}}) & R_2 < ST \\ X_{i,j}^t + Q \cdot L & R_2 \geq ST \end{cases} \tag{21}$$

where $X_{i,j}^{t+1}$ denotes the position information of the $i$ sparrow in the $j$ dimension under $t+1$ iteration; $iter_{\max}$ denotes the maximum number of iterations, which is a constant; $R_2$ and $ST$ denote the warning value and the safety value, respectively; $\alpha \in (0,1]$; $Q$ denotes the random number obeying the normal distribution; and $L$ denotes the matrix of dimension $1 \times d$, each element of which is 1.

When $R_2 < ST$, there is no danger in the foraging area and the explorers can forage in a large area; when $R_2 \geq ST$, there is danger and an alarm is issued, and then all sparrows move to the safe area to forage.

By rule 3 and rule 4, when the followers perceive that the explorers have acquired better food, they will leave their existing position to compete and, if the followers win, they will immediately get the physical object; otherwise, they will update their position according to Equation (22)

$$X_{i,j}^{t+1} = \begin{cases} Q \cdot \exp(\frac{X_{worst}^t - X_{i,j}^t}{i^2}) & i > \frac{n}{2} \\ X_p^{t+1} + \left| X_{i,j}^t - X_p^{t+1} \right| \cdot A^+ \cdot L & others \end{cases} \tag{22}$$

where $X_{worst}^t$ denotes the global worst position at $t$ iteration; $X_p^{t+1}$ denotes the optimal position occupied by the discoverer at $t+1$ iteration; and $A^+ = A^T(AA^T)^{-1}$, $A$ denotes a matrix of dimension $1 \times d$, whose elements are 1 and $-1$. When $i > n/2$, the first joiner with low fitness does not get food and needs to fly to other areas for food.

The number of sparrows aware of the danger is set to 20% of the total population in the experiment. At the same time, according to rules 5 and 6, the initial positions of these sparrows are randomly generated with the mathematical expression

$$X_{i,j}^{t+1} = \begin{cases} X_{best}^t + \beta \left| X_{i,j}^t - X_{best}^t \right| & f_i > f_g \\ X_{i,j}^t + K \cdot (\frac{\left| X_{i,j}^t - X_{worst}^t \right|}{(f_i - f_w) + \varepsilon}) & f_i = f_g \end{cases} \tag{23}$$

where $X_{best}^t$ denotes the global optimal position at $t$ iteration; $\beta$ is the step control parameter, and $\beta \sim N(0,1)$, $K$ denotes the direction of sparrow movement; $f_i$, $f_g$ and $f_w$ denote the current sparrow individual, global best and global worst fitness values, respectively; and $\varepsilon$ is a constant to ensure that the denominator is not zero.

When $f_i > f_g$, the sparrows at the edge of the population are vulnerable to attack; when $f_i = f_g$, the sparrows in the middle of the population are aware of the danger and start to approach other sparrows.

### 3.3. Localization Method for FDIAs Based on the SSA–CNN

The structure of the localization method for FDIAs proposed in this paper is shown in Figure 2. The input is a dimensional measurement vector, each row of which represents a measurement vector. The value of each dimension of each measurement vector represents different specific physical quantities in the power network, namely active power, reactive power, line active current and line reactive current from Equation (2). The measurement vectors are classified with the SSA–CNN classification model proposed in this paper, where each dimension of the measurement vectors represents a label. After the classification model, the output classification vectors are also dimensional vectors, but the elements in the classification vectors are all 0 and 1 because, after the measurement vectors pass through the last layer of the CNN that is the fully connected layer, the data can be mapped

to the interval $(0, 1)$ due to the characteristics of the sigmoid activation function; then, the mapped values are classified as 0 and 1 with the logical judgment formula, which constitute the output classification vectors.

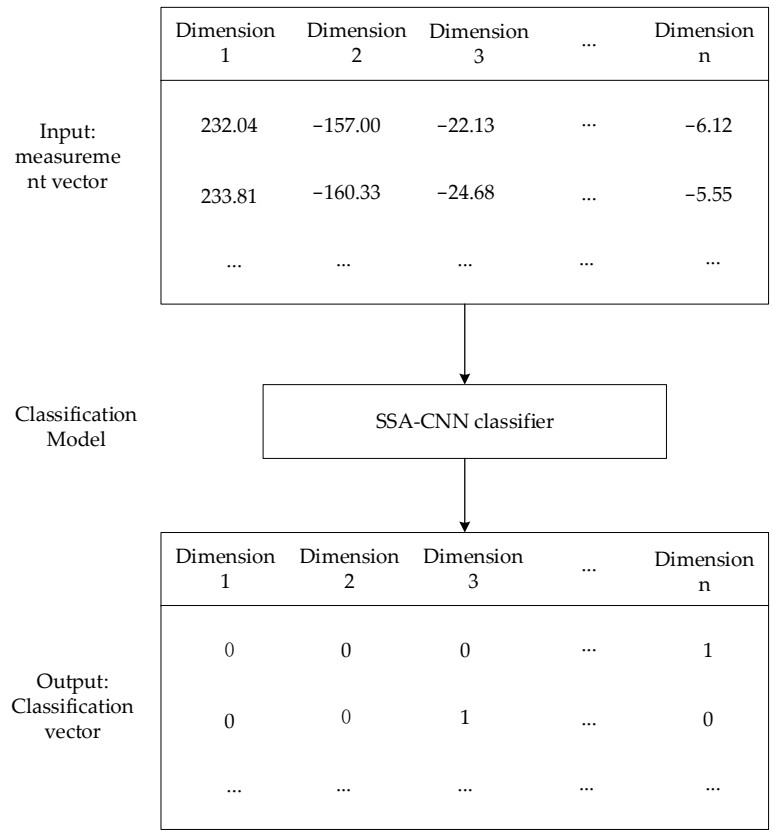

**Figure 2.** Structure of the localization method for FDIAs.

The value of each dimension of the output classification vectors is used to determine whether a specific bus or line is under a false data injection attack. As in Figure 2, for the 2nd line measurement vector $(233.81, -160.33, -24.68, \cdots, -5.55)^T$, the data in the first three dimensions represent the active power of bus 1, the active current flowing from bus 2 to bus 1 and the active current flowing from bus 3 to bus 4, respectively. The output classification vector obtained is $(0, 0, 1, \cdots, 0)^T$; therefore, based on the classification vector result, the lines between bus 3 and bus 4 can be judged to be under attack. In fact, attackers do not just attack a bus or a line, they usually attack a small area, which may contain several buses and several lines, so the output classification vector becomes a one-dimensional column vector consisting of several ones and zeros.

## 4. Simulation Experiments

### 4.1. Simulation Experiment Settings

The simulation experiments were performed on IEEE14-bus and IEEE118-bus test systems. In order to realize the localization method for FDIAs, information, such as the topology and parameters of the test systems, needs to be obtained.

Taking the IEEE14-bus test system as an example, its topology and parameter information are obtained from MATPOWER, whose system information is shown in Table 1, and its topology and measurement information are shown in Figure 3. In Figure 3, the numbers alone represent different buses, the numbers in the circles represent different flow measurements, and the numbers in the rectangles represent different injection measurements.

**Table 1.** Information of the IEEE14-bus Test System.

| Parameter | Number |
|---|---|
| Buses | 14 |
| Lines | 20 |
| Total Measurements | 19 |
| Injection Measurements | 7 |
| Flow Measurements | 12 |
| Unmeasured Lines | 8 |

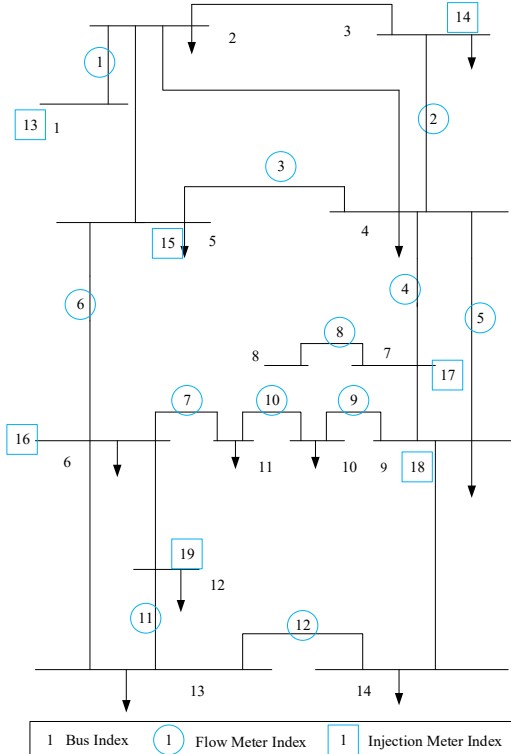

**Figure 3.** IEEE14-bus Test System.

The IEEE14-bus test system has 19 measurements, of which seven are injection measurements, and 12 are flow measurements, as shown in Figure 3. The flow between bus 1 and bus 2 is measured by meter 1, injected power at bus 1 is measured by meter 13, and the rest are similar.

Since a normal FDIA can be detected with the smart grids' bad data detectors, this paper uses the construction of invisible false data as described in Section 2.2 to construct the attack vector. Taking the IEEE14-bus system as an example, there are 19 dimensions of measurement vectors. The true values of the measurement vectors are obtained from MATPOWER, adding Gaussian noise with a variance of 0.02 as the actual measurement vectors. Then, the attack vectors are added to the actual measurement vectors to generate the measurement vectors being attacked. Only parts of the areas are hit, so only the corresponding parts of the measurement vectors are tampered with. According to the above method, the measurement vectors being attacked are generated randomly; at the same time, the corresponding classification vector sets are generated. For the IEEE14-bus test system, 12,000 volume measurement vectors are generated, and the ratio of the training set, validation set and test set is 4:1:1; for the IEEE118-bus test system, 6000 volume measurement vectors are generated, and the ratio of the training set, validation set and test set is also 4:1:1.

To evaluate the localization accuracy of the algorithms, the true positive rate ($TPR$), false positive rate ($FPR$) and $F1$ score are used as the performance evaluation metrics. In

this paper, the true positive rate $TPR$ is defined as the probability that the attacked region is successfully located, the false positive rate $FPR$ indicates the probability that the region not attacked is mislocated, and the $F1$ score is a composite metric that indicates the balance between the accuracy rate and the recall rate. The expressions of $TPR$, $FPR$, *Precision*, *Recall* and $F1$ score are shown as

$$TPR = \frac{TP}{TP + FN} \tag{24}$$

$$FPR = \frac{FP}{FP + TN} \tag{25}$$

$$Precision = \frac{TP}{TP + FP} \tag{26}$$

$$Recall = \frac{TP}{TP + FN} \tag{27}$$

$$F1 = 2 \times \frac{Precision \times Recall}{Precision + Recall} \tag{28}$$

where $TP$ denotes the number of successfully located attacked localizations, $FN$ denotes the number of attacked localizations that were not located, $FP$ denotes the number of regions not attacked that were mislocated, and $TN$ denotes the number of regions not attacked that were not located.

### 4.2. Simulation Results of the SSA–CNN

For the CNN model, there are six hyperparameters to be optimized, which are learning rate, number of iterations, batch size, number of filters, length of convolutional kernels and number of convolutional layers. Together, these six hyperparameters constitute the population number in the SSA. Table 2 shows the optimization range of the SSA for different hyperparameters and the optimization results in the two test systems, IEEE14-bus and IEEE118-bus. For different systems, the SSA optimization results for all six hyperparameters are different, which is caused by the system size and data differences.

**Table 2.** SSA optimized hyperparameters of the CNN model.

| Model | Hyperparameter | Optimization Scope | Optimization Results | |
| --- | --- | --- | --- | --- |
| | | | IEEE-14 | IEEE-118 |
| | learning rate | [0.001, 0.01] | 0.0036 | 0.0020 |
| | number of iterations | [50, 200] | 83 | 175 |
| CNN | batch size | [100, 200] | 190 | 157 |
| | number of filters | {16, 32, 64, 128, 256} | 64 | 128 |
| | length of convolutional kernels | [2, 5] | 3 | 2 |
| | number of convolutional layers | [2, 5] | 3 | 4 |

The maximum number of iterations M of the SSA is set to 10, and the adaptation value change curves of the SSA for the IEEE14-bus and IEEE118-bus test systems are shown in Figures 4 and 5, respectively, where the number of iterations is calculated from 0. It can be observed from the figures that, in the IEEE14-bus test system, the fitness value decreases rapidly as the number of iterations increases; after the 5th iteration, the decline begins to slow down and stabilizes by the 7th iteration; and finally, the fitness value at the end of the 7th, 8th and 9th iterations is the same.

In the IEEE118-bus test system, the fitness value decreases more rapidly from the 0th to the 2nd iteration, after which the fitness value decreases more slowly and eventually stabilizes after the 6th iteration with the same value at the end of the 6th, 7th, 8th and 9th iterations.

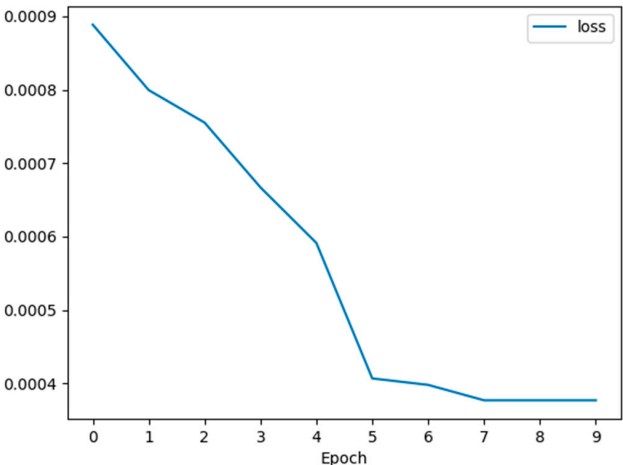

**Figure 4.** Adaptability values of the SSA for the IEEE14-bus test system.

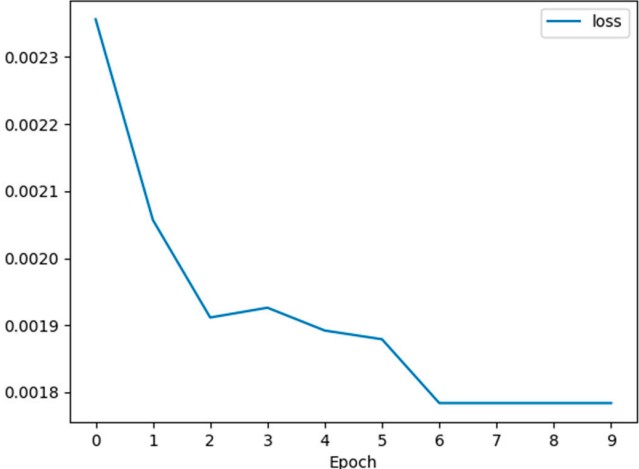

**Figure 5.** Adaptability values of the SSA for the IEEE118-bus test system.

Table 3 shows the fitness values of the IEEE14-bus and IEEE118-bus test systems for each iteration. It can be observed that the fitness value of the IEEE14-bus test system converges to 0.00038 after the 7th iteration, and that of the IEEE118-bus test system converges to 0.00178 after the 6th iteration. This may be due to the fact that the CNN model is more fitting and adaptive to the data of the IEEE14-bus test system, which will also lead to a more efficient and effective localization of the SSA–CNN algorithms on the IEEE14-bus test system.

**Table 3.** Iterative adaptation values for the IEEE14-bus and IEEE118-bus test systems.

| Number of Iterations | IEEE14 | IEEE118 |
|---|---|---|
| 0 | 0.00089 | 0.00236 |
| 1 | 0.00080 | 0.00206 |
| 2 | 0.00076 | 0.00191 |
| 3 | 0.00067 | 0.00193 |
| 4 | 0.00059 | 0.00189 |
| 5 | 0.00041 | 0.00188 |
| 6 | 0.00040 | 0.00178 |
| 7 | 0.00038 | 0.00178 |
| 8 | 0.00038 | 0.00178 |
| 9 | 0.00038 | 0.00178 |

The localization simulation experiments of the FDIA are performed on the IEEE14-bus and IEEE118-bus test systems using the optimized hyperparameters of the SSA in Table 2. Figures 6 and 7 show the change curves of accuracy and loss value of the CNN model during training. As can be observed from the figures, both accuracy and loss value converge quickly. Although there are some fluctuations in the training process, they tend to stabilize.

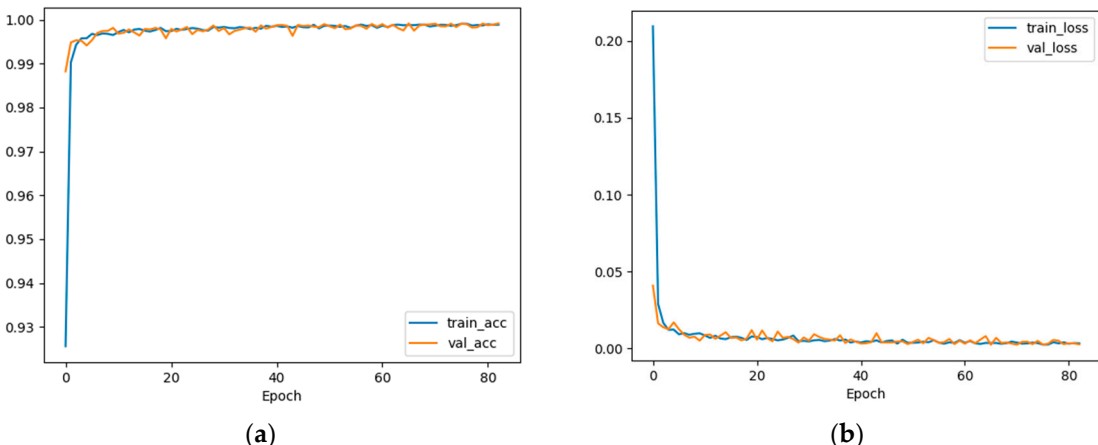

(a)                                    (b)

**Figure 6.** Accuracy and loss value curves of the IEEE14-bus test system. (**a**) Accuracy value curves of the IEEE14-bus test system. (**b**) Loss value curves of the IEEE14-bus test system.

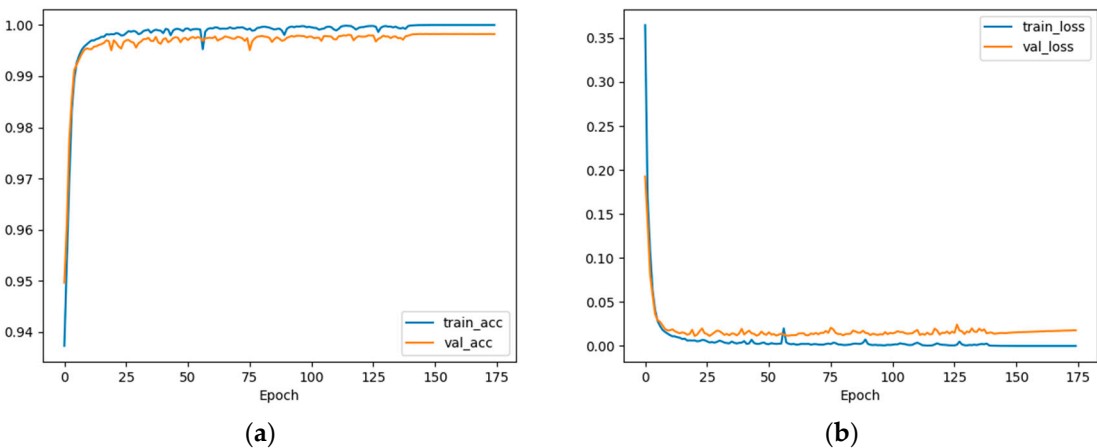

(a)                                    (b)

**Figure 7.** Accuracy and loss value curves of the IEEE118-bus test system. (**a**) Accuracy value curves of the IEEE118-bus test system. (**b**) Loss value curves of the IEEE118-bus test system.

The SSA–CNN localization algorithms proposed in this paper are simulated and compared with the common CNN, the DNN localization algorithms in [20], the DT localization algorithms in [29], the KNN localization algorithms in [30] and the ELM localization algorithms in [31]. To be fair, all localization algorithms are performed using the same data sets. In Table 4, the performance metrics of the different localization algorithms in the two tested systems are presented in percentage form. For the IEEE14-bus test system, the TPR, FPR and F1 scores of the algorithms proposed in this paper are 99.85, 0.03 and 99.89, respectively, which are the best among all the algorithms. The TPR and F1 scores of the CNN, DNN and ELM are slightly lower than those of this paper, but their FPR is higher than that of this paper with the FPR of the CNN being 53 times higher than that of this paper. The $TPR$ of DT and KNN is not satisfactory, only 75.37 and 79.78, respectively; at the same time, the FPR is also higher. For the IEEE118-bus test system, the $TPR$, $FPR$ and $F1$ scores of this paper's algorithms are 97.14, 0.03 and 98.27, respectively, and the $TPR$ is slightly lower than that of the IEEE14-bus test system, which may be due to the increase

in the measurement data and noise. The $TPR$ of the CNN is 8% lower than that of the IEEE14-bus test system; in contrast, the DNN's $TPR$ increases slightly, but its FPR increases rapidly, leading to a decrease in its F1 score. The $TPR$ of ELM in the IEEE118-bus test system is 88.78%, and the $FPR$ is higher than that of the IEEE14-bus test system, which is 2.40%. The performance index of the DT localization algorithms is similar to that of the IEEE14-bus test system, but the $TPR$ of the KNN localization algorithms decreases sharply, which leads to a sharp decrease in its $F1$ score.

**Table 4.** Simulation results of the different algorithms.

| Performance Index | IEEE-14 | | | IEEE-118 | | |
|---|---|---|---|---|---|---|
| | *TPR* | *FPR* | *F1* | *TPR* | *FPR* | *F1* |
| SSA–CNN | 99.85 | 0.03 | 99.89 | 97.14 | 0.03 | 98.27 |
| CNN | 97.34 | 1.59 | 96.99 | 89.34 | 0.23 | 92.19 |
| DNN | 92.95 | 0.43 | 95.89 | 93.01 | 2.56 | 93.72 |
| DT | 75.37 | 1.18 | 77.00 | 72.31 | 1.33 | 73.18 |
| KNN | 79.78 | 0.94 | 87.78 | 56.61 | 1.12 | 63.60 |
| ELM | 93.79 | 1.34 | 95.40 | 88.78 | 2.40 | 91.58 |

In summary, the SSA–CNN localization algorithms proposed in this paper have better performance indicators than other localization algorithms in terms of the $TPR$, $FPR$ and the overall $F1$ score. It should be emphasized that the localization algorithms for FDIAs in this paper locate the attack to a specific bus or line, not to a certain area, and the simulation and performance indexes are implemented and calculated accordingly.

## 5. Discussion

As a kind of cyber–physical system, a smart grid is vulnerable to network attacks, among which, the FDIA is the typical one with strong concealment and destructiveness. Through the method proposed in this paper, we can efficiently locate the false data injection attack so as to guarantee the safe and stable operation of smart grids.

## 6. Conclusions

In this manuscript, localization algorithms for FDIAs based on the SSA–CNN for smart grids are proposed. The measurement equations of smart grids and the way of constructing an invisible FDIA are described in the second chapter. In the third chapter, the CNN model, SSA and localization algorithms for the localization of FDIAs are proposed that use the SSA–CNN to classify the measurement data of smart grids and then localize the attack according to the physical significance of each measurement data, which can pinpoint the specific attacked bus or line rather than the attack area. In the fourth chapter, the proposed algorithms are simulated in the IEEE14-bus and IEEE118-bus test systems; the results show that the algorithms have 99.85% and 97.14% localization accuracy in the two systems, respectively, and the false detection rate is only 0.03% in both systems. The comparison of the proposed algorithms with other advanced attack localization algorithms shows that the localization accuracy of the other algorithms is lower than that of the proposed algorithms, which proves the superiority of the proposed algorithms in the localization of FDIAs.

With the progress of automatic control and communication technologies, there are expanding types of network attacks, and the attack methods are becoming increasingly complex. The next stage of research focuses on the applicability of the algorithms to different network attacks and the effectiveness for new types of network attacks.

**Author Contributions:** Conceptualization, K.S. and W.Y.; methodology, K.S.; software, K.S.; validation, K.S. and W.Y.; formal analysis, K.S.; investigation, K.S.; resources, K.S.; data curation, K.S.; writing—original draft preparation, K.S.; writing—review and editing, W.Y.; visualization, K.S.; supervision, W.Y.; project administration, W.Y.; funding acquisition, H.N. and J.C. All authors have read and agreed to the published version of the manuscript.

**Funding:** This research was funded by the State Grid Zhejiang Province Technology Project, grant number 5211SX220003.

**Institutional Review Board Statement:** Not applicable.

**Informed Consent Statement:** Not applicable.

**Data Availability Statement:** Not applicable.

**Conflicts of Interest:** The authors declare no conflict of interest.

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
