# Peer review of "Localization of False Data Injection Attack in Smart Grids Based on SSA-CNN"

_information, doi:10.3390/info14030180_

Round 1

Reviewer 1 Report

This paper proposes a method based on sparrow search algorithm optimized convolutional neural network (SSA-CNN) for localization of FDIA in smart grid. However, some descriptions are not clear. Some revisions are necessary in the manuscript.

1. SSA-CNN is not a novel approach, please explain in detail what innovative work has been done to the algorithm.

2. This article describes SSA and CNN separately, please explain how the two are combined.

3. Please explain how the simulation parameters are designed.

4. Smart grid is mentioned in the article. Please further explain the impact of FDIA on smart grid.

5. In the paper, authors have focused on the data-driven approach for complex power systems. Comparison of different data-driven approaches for complex power systems needs to be analyzed to indicate advantages of your work, which can refer to:

[a] IEEE Transactions on Network Science and Engineering, vol. 9, no. 4, pp. 2301-2316, 1 July-Aug. 2022.

[b] IEEE Trans. Power Systems, vol. 34, no. 6, pp. 4557-4568, 2019

[c] IEEE Trans. Ind. Inf. DOI: 10.1109/TII.2023.3241682

[d] IEEE Trans. Power Systems, vol. 35, no. 1, pp. 731-741, 2020

Author Response

All responses to the reviewer 1’s comments are in the attachment, please see the attachment.

Reviewer 2 Report

In this manuscript, an SSA-CNN based FDIA localization algorithm for smart grid is proposed. Measurement equations of smart grid and the way of constructing invisible FDIA are described in the second chapter. In the third chapter, the CNN model, optimization algorithm and FDIA localization algorithm for FDIA localization are proposed. Using SSA-CNN to classify the measurement data of smart grid, and then localize the attack 468 according to the physical significance of each measurement data, which can pinpoint to the specific attacked bus or line rather than attack area. In the fourth chapter, the proposed algorithm is simulated in IEEE14-bus and IEEE118-bus systems, the results show that the algorithm has 99.85% and 97.14% localization accuracy in the two systems, respectively, and the false detection rate is only 0.03% in both systems.

I recommend the following minor revisions.

1. The introduction section contains lengthy paragraphs. Please split them into multiple paragraphs.

2. Please specify the main contribution of the paper in bullet or numbered form.

3. Please include some future scope of the works in the end of conclusion section.

4. The paper needs a thorough proof read.

Author Response

All responses to the reviewer 2’s comments are in the attachment, please see the attachment.

Reviewer 3 Report

-          The quality of figures 6 and 7 need to be improved. At their current forms, it is hard to distinguish among different lines.

-          The word “Grid” in the title should be plural.

-          In the first sentence of the abstract the it seems that it should be “smart grids” not “the smart grid”.

-          From this reviewer’s point of view, the first sentence of the abstract is not thoroughly correct. I mean integration of information systems into power systems have not enabled FDIA. It provides the situation for these attacks to be happened. Moreover, the integration of “physical systems” into smart grids seems unclear for me.

-          Since these types of models have to be applied on large power networks, please add an explanation on the complexity order of the proposed model.

-          Please highlight the contributions of the paper at the end of Introduction section.

-          Line 137: the verb of this sentences should be “are”.

-          Line 149: “algorithm is” should be modified to “algorithms are”. All of the paper should be revised form this point of view. All of this type of errors have to be corrected.  

-          It will be better if you use different symbols for different types of meters in figure 3.

-          I could not understand how the training sets are created? It needs to be expalied with more details. 

Author Response

All responses to the reviewer 3’s comments are in the attachment, please see the attachment.

Round 2

Reviewer 3 Report

All of the comments have been addressed.